# ResMiCo: Increasing the quality of metagenome-assembled genomes with deep learning

Olga Mineeva[1,2,3‡], Daniel Danciu[1‡], Bernhard Schölkopf[1,2,4], Ruth E. Ley[5], Gunnar Rätsch[1,3,4,6,7]*, Nicholas D. Youngblut[5]*

1 Department of Computer Science, ETH Zürich, Zürich, Switzerland, 2 Department of Empirical Inference, Max Planck Institute for Intelligent Systems, Tübingen, Germany, 3 Swiss Institute for Bioinformatics, Lausanne, Switzerland, 4 ETH AI center, ETH Zürich, Zürich, Switzerland, 5 Department of Microbiome Science, Max Planck Institute for Biology, Tübingen, Germany, 6 Department of Biology, ETH Zürich, Zürich, Switzerland, 7 Medical Informatics Unit, Zürich University Hospital, Zürich, Switzerland

‡ These authors share first authorship on this work.
* gunnar.raetsch@inf.ethz.ch (GR); nyoungblut@tuebingen.mpg.de (NDY)

**Data Availability Statement:** Resmico is publicly available at https://github.com/leylabmpi/ResMiCo. The computationally intensive feature extraction library was written in C++, and the data simulation

## Abstract

The number of published metagenome assemblies is rapidly growing due to advances in sequencing technologies. However, sequencing errors, variable coverage, repetitive genomic regions, and other factors can produce misassemblies, which are challenging to detect for taxonomically novel genomic data. Assembly errors can affect all downstream analyses of the assemblies. Accuracy for the state of the art in reference-free misassembly prediction does not exceed an AUPRC of 0.57, and it is not clear how well these models generalize to real-world data. Here, we present the Residual neural network for Misassembled Contig identification (ResMiCo), a deep learning approach for reference-free identification of misassembled contigs. To develop ResMiCo, we first generated a training dataset of unprecedented size and complexity that can be used for further benchmarking and developments in the field. Through rigorous validation, we show that ResMiCo is substantially more accurate than the state of the art, and the model is robust to novel taxonomic diversity and varying assembly methods. ResMiCo estimated 7% misassembled contigs per metagenome across multiple real-world datasets. We demonstrate how ResMiCo can be used to optimize metagenome assembly hyperparameters to improve accuracy, instead of optimizing solely for contiguity. The accuracy, robustness, and ease-of-use of ResMiCo make the tool suitable for general quality control of metagenome assemblies and assembly methodology optimization.

## Author summary

Metagenome assembly quality is fundamental to all downstream analyses of such data. The number of metagenome assemblies, especially metagenome-assembled genomes (MAGs), is rapidly increasing, but tools to assess the quality of these assemblies lack the accuracy needed for robust quality control. Moreover, existing models have been trained

pipeline was implemented using Snakemake https://doi.org/10.1093/bioinformatics/bts480 [47]. The deep learning model was built using Tensorflow http://download.tensorflow.org/paper/whitepaper2015.pdf [48]. MGSIM is available at https://github.com/nick-youngblut/MGSIM. Due to the size of the n9k-train and n9k-novel datasets, these data has been deposited on the MPI for Biology FTP server and can be downloaded with tools such as wget or curl through the following link: http://ftp.tue.mpg.de/ebio/projects/ResMiCo/. The authors are committed to ensuring the perpetual public accessibility of these data.

**Funding:** This work was supported by Eidgenössische Technische Hochschule Zürich core funding (OM, DD, and GR), the Max Planck Institute (NDY, REL, and BS), and the Eidgenössische Technische Hochschule Strategic Focus Area - Personalized Health and Related Technologies (project #106 to DD). OM was also supported by the Max Planck ETH Center for Learning Systems. The funders had no role in study design, data collection and analysis, decision to publish, or preparation of the manuscript. OM, DD, and GR received a salary from the Eidgenössische Technische Hochschule Zürich; NDY, REL, and BS received a salary from the Max Planck Institute.

**Competing interests:** The authors have declared that no competing interests exist.

on datasets lacking complexity and realism, which may limit their generalization to novel data. Due to the limitations of existing models, most studies forgo such approaches and instead rely on CheckM to assess assembly quality, an approach that only utilizes a small portion of all genomic information and does not identify specific misassemblies. We harnessed existing large genomic datasets and high-performance computing to produce a training dataset of unprecedented size and complexity and thereby trained a deep learning model for predicting misassemblies that can robustly generalize to novel taxonomy and varying assembly methodologies.

This is a *PLOS Computational Biology* Methods paper.

## Introduction

Metagenome sequencing is rapidly increasing in popularity due to the lowering costs of sequencing and simplified library construction methods [1, 2]. At the same time, improvements in metagenome assembly tools [3, 4] and high-performance computing resources have increased the feasibility of large-scale metagenome assemblies on 1000s of samples [5–7]. The contiguous sequences (contigs) generated via metagenome assembly can be analyzed directly for such tasks as creating gene catalogs [8, 9], or binning approaches can be used to cluster the contigs into metagenome-assembled genomes (MAGs) that can be used for various comparative genomics applications [10].

These advances have given rise to vast genome assembly databases, such as the Unified Human Gastrointestinal Genome (UHGG) [11], in which MAGs account for 70% of the species. As another example, the Genome Taxonomy Database (GTDB) expanded from approximately 32,000 species to nearly 50,000 in less than one year [12, 13], largely due to the proliferation of MAGs. Given the low-throughput nature of isolating Bacteria and Archaea for independent genome sequencing [11], metagenome assembly approaches will likely continue to dominate.

The correct assembly of metagenomes is challenging due to several factors, including sequencing errors, high taxonomic diversity often comprising 1000s of species, uneven coverage, and repetitive genomic regions [14]. All of these factors contribute to misassemblies, with the most common being structural variations, relocations, translocations, and inversions [15]. Long read sequence data can mitigate some of these issues [14, 16], but the expense relative to short read sequencing generally prevents one from obtaining sufficient sequence coverage for complex communities [17]. While assembly contiguity can be assessed easily by calculating such metrics as N50, assessing assembly accuracy is considerably more challenging due to a few major causes. First, due to a lack of very closely related taxa with genome regions (nearly) identical to the query, contigs cannot simply be mapped to references in order to assess accuracy. Second, reference-free tools that predict misassemblies have generally been trained and validated on small, homogeneous datasets in the past, which raises the question of their robustness to novel data (e.g., novel taxa or assembly methods). Indeed, Mineeva and colleagues showed that existing tools generally performed poorly on a large, heterogeneous dataset [18]. The authors' novel deep learning approach, DeepMAsED, amply outperformed the state of the art and was relatively robust to taxonomic novelty, achieving an AUPRC score of 0.57 on a

novel genome dataset; still, there remained substantial room for improvement in model accuracy and also a robust validation on complex datasets spanning the heterogeneity of existing metagenomes from complex communities. Indeed, Lei and colleagues developed metaMIC, a reference-free machine learning (ML) model for predicting misassemblies in metagenome assemblies [19], and showed that DeepMAsED's performance was inferior to metaMIC's; however, only few methodological details were provided on the validation approach.

We present Residual Neural Network for Misassembled Contigs Identification (ResMiCo), a novel approach for reference-free identification of misassembled contigs in metagenome assemblies. ResMiCo is a deep convolutional neural network with skip connections between non-adjacent layers. Similar architectures have proven to be highly successful when trained on large datasets from various fields [20, 21]. We utilize a novel high throughput pipeline to generate complex and realistic training data covering much of the possible parameter space (e.g., varying data richness, sequencing depth, sequencing error rate, community diversities, and metagenome assembly methods). Through extensive evaluation, we show that the model outperforms the existing state of the art and is robust to metagenome data heterogeneity, including taxonomic novelty and metagenome assembly parameters. ResMiCo is also robust to alternative data simulation approaches, as shown when applied to the Critical Assessment Metagenome Interpretation (CAMI) datasets. We show that using ResMiCo to filter putative genomes reduces the number of misassembled contigs by a factor of four. We also apply ResMiCo to a large collection of metagenomes from published studies and show that 7% ± 5.7 (s.d.) of contigs per metagenome are misassembled. Lastly, we show that ResMiCo can be used to optimize metagenome assembler parameters for accuracy without the need for simulated or mock-community metagenome datasets.

## Materials and methods

### Simulated data

We used synthetic datasets for initial model training and testing. The data simulation methodology, depicted in Fig 1, builds on, and significantly expands on our previous work [18]. Reference bacterial and archaeal genomes were selected from Release 202 of the Genome Taxonomy Database (GTDB) [22]. Metagenomes were simulated from publicly available reference genomes via MGSIM (https://github.com/nick-youngblut/MGSIM). Simulation parameters varied in all combinations of i) community richness, ii) community abundance distribution, iii) reference genomes selected from the total pool, iv) read length, v) insert size distribution (i.e., the distance in between the forward and reverse read pair), vi) sequencer error profile, vii) sequencing depth, and viii) metagenome assembler (Table 1). The abundance distribution of each community was modeled as a log-normal distribution. We varied parameter $\sigma$ to produce differing levels of evenness of relative abundances. Community richness was altered by random sub-sampling from the pool of reference genomes available in the training or test split. The ART [23] read simulator was used to generate paired-end Illumina reads of length 100 or 150 using either the default "Illumina HiSeq 2500" error profile or the "HiSeq2500L150R1/2" error profile used in CAMISIM [24]. Four paired-end read insert size distributions were simulated via the ART parameter settings (see "Insert size" in Table 1). We included multiple simulation replicates, in which the community and read simulation parameters were held constant, but each replicate differed via randomization of the genome sub-sampling within each simulation. The reads from each community were assembled independently with metaSPAdes [3] and MEGAHIT [4].

MetaQUAST [15] was used to identify truly misassembled contigs based on mapping all contigs to the reference genomes used for the simulations. The MetaQUAST-identified contig

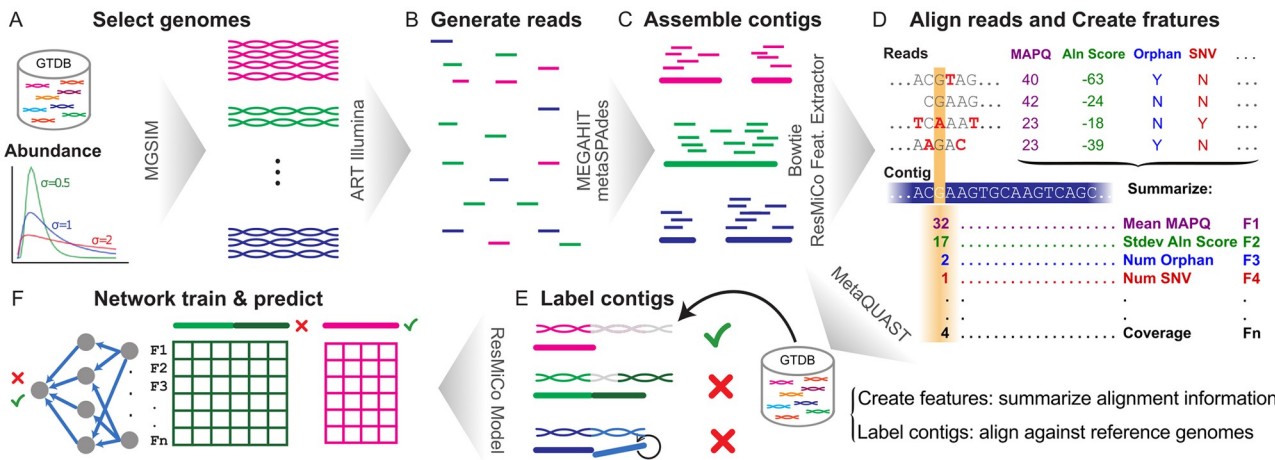

**Fig 1. The ResMiCo simulation and training pipeline.** (A) Select reference genomes from the Genome Taxonomy Database (GTDB) at various abundances; (B) Simulate reads for the selected genomes using ART-Illumina; (C) Assemble reads into contigs using MEGAHIT and metaSPAdes; (D) Align reads back to the assembled contigs using Bowtie2, then extract features such as coverage, number of single-nucleotide variants (SNVs), mean alignment score, etc., for each contig using the given alignments; (E) Compute labels for each contig by aligning against the reference genomes using MetaQUAST; (F) For each contig, select a random section (that contains a breakpoint if the contig is misassembled), pad to the network's input length if necessary, and feed the data into the ResMiCo model. Steps (D) and (E) are independent and can be parallelized.

misassembly labels ("extensive misassembly") were used as ground truth. The extensive misassembly label includes relocations, inversions, translocations and interspecies translocations. We also extracted MetaQUAST-identified breakpoint positions in the misassembled contigs.

For initial model training and testing, we utilized a pool of 18,000 reference genomes selected from Release 202 of the Genome Taxonomy Database (GTDB). The pool was split at the family taxonomic level so that all genomes in the test dataset belonged to families not present in the training dataset. The resulting split was even, with 9000 genomes used for both training and testing. To reduce bias toward particular species, at most 50 genomes per species were included in the reference genome pool, with genomes selected at random. The pool was also filtered by CheckM-estimated completeness ($\geq$ 90%) and contamination ($\leq$ 5%). Other filtering criteria included: i) only "high" MIMAG quality, ii) no single-cell genome assemblies,

**Table 1. Parameter values used in the simulation pipeline.** The training dataset *n9k-train* was generated using all 1440 parameter combinations for the insert size distribution with mean = 270 & sd = 50 and mean = 190 & sd = 75. For the "mean = 350 & sd = 75" insert size distribution, 3 replicates for the "HiSeq 2500 L150R1/2" error profile and one replicate for the "HiSeq 2500 error" profile were generated (960 parameter combinations). For the "mean = 450 & sd = 120" insert size distribution, we used all parameters combinations, but only with the "HiSeq 2500 L150R1/2" error profile (720 parameter combinations). In total, we created 4560 samples. The parameter grid was not uniformly sampled due to the resource limitations. The test dataset *n9k-novel* contains one simulation replicate with the "HiSeq 2500" error profile, and using one (mean = 270 & sd = 50) insert size, which resulted in 240 parameter combinations.

| Parameter | Values |
|---|---|
| Community richness | 50, 1000, 3000, 5000 genomes |
| Genome abundance | Lognormal with $\mu$ = 10 and $\sigma \in$ {0.5, 1, 2} |
| Replicates | 3 random selections of reference genomes |
| Read length | 100, 150 bps |
| Insert size | mean & sd $\in$ {190&75, 270&50, 350&75, 450&120} |
| Error profile | HiSeq 2500, HiSeq 2500 L150R1/2 [24] |
| Sequencing depth | 0.5m, 2m, 8m, 12m, 20m |
| Assembler | MEGAHIT, MetaSPAdes |

**Table 2. A summary of the five synthetic datasets used for training and evaluating ResMiCo.** The *n9k-train* and *n9k-novel* datasets were generated using our pipeline, all other datasets were created from CAMI reads. The *n9k-train* dataset was used for training and validation, while all other datasets were used for testing. Misassemblies are reported as a percent of the total number of contigs in the dataset. Misassemblies length is the sum of misassembled contig lengths divided by the total number of bases.

|  | n9k-train | n9k-novel | gut | skin | oral | marine | plant |
|---|---|---|---|---|---|---|---|
| Contigs | 52.5M | 6.8M | 0.44M | 0.32M | .41M | 1.0 | 1.0 |
| Bases | 159B | 19.7B | 1.5B | 890M | 1.5B | 2.3B | 3.4B |
| Average Coverage | 12.6 | 10.2 | 16.9 | 15.7 | 16.3 | 14.1 | 16.7 |
| Misassemblies | 3.41% | 4.0% | 1.2% | 1.72% | 1.74% | 2.66% | 2.43% |
| Misassemblies length | 3.69% | 4.26% | 3.82% | 3.06% | 3.20% | 3.29% | 6.70% |
| Median Contig length | 1530 | 1510 | 1473 | 1394 | 1455 | 1349 | 1427 |

iii) $\leq$ 500 contigs, iv) a genome size of $\leq$ 15 mbp, and v) a mean contig length of $\geq$ 10 kbp. We randomly subsampled reads mapped to each contig to a maximum mean contig coverage of 20. At this coverage level, assemblies of reasonable quality can be produced, and subsampling helps prevent out-of-distribution issues when applying ResMiCo to datasets with substantially higher sequencing depth than in our training dataset.

To create features for model training and testing, we mapped reads to the contigs from the corresponding synthetic metagenome via Bowtie2 [25], and the resulting alignment data was used to generate per contig position features, as listed in Table A in S1 Text.

We refer to the training dataset as *n9k-train* and to the test dataset, which consists of genomes novel at a family taxonomic level, as *n9k-novel*. All combinations of the metagenome simulation parameters produced a training dataset of 4560 metagenomes and 80M contigs (Table 1). We were limited to 1.3T of available space on our HPC cluster, so we had to randomly subset *n9k-train* to 3000 metagenomes, comprising 52M contigs (Table 2). The test set (*n9k-novel*) was generated using the subset of parameters and one simulation replicate to save computational time (Table 1). We also created a test dataset with higher intra-species diversity (33.3 ± 115) versus *n9k-novel* (2.4 ± 13.7), which we named *n2k-novel-intra-species*. We used only 2000 reference genomes in order to include enough families with a high number of intra-species genome diversity, but still include only families novel relative to the train dataset (as with *n9k-novel*). The simulation parameters for *n2k-novel-intra-species* were consistent with *n9k-train*, except i) only 50 and 1000 genomes for richness, ii) one simulation replicate, iii) one insert size distribution (mean = 270, sd = 50), and iv) sequencing depths of 2M, 8M, and 12M read pairs.

## CAMI simulated metagenomes

To benchmark ResMiCo's performance in a new setting, we downloaded the paired-end reads of the Critical Assessment of Metagenome Interpretation (CAMI) human skin, human oral, human gut, plant-associated, and marine assembly challenges [26]. As with the *n9k-train* dataset, we assembled the reads via metaSPAdes and MEGAHIT, and identified true misassemblies via MetaQUAST based on the reference genomes in each of the 5 datasets. As shown in Table 2, the number of misassembled contigs in the CAMI datasets is ∼50% lower, while coverage is ∼50% higher relative to the *n9k-train* dataset. The breakpoint locations for misassembled contigs follow a nearly identical distribution for all datasets, with more breakpoints clustered towards the ends (Fig A in S1 Text).

## Published, real-world metagenomes

We evaluated ResMiCo on 7 published metagenome datasets: *UHGG* [11], *TwinsUK* [27], *Animal-gut* [28], *Pinnell2019* [29], *Mantri2021* [30], *MarineMetagenomeDB* [31], and

*TerrestrialMetagenomeDB* [32] (S1 Table). The *UHGG* consisted of randomly selected metagenomes associated with the UHGG MAG collection. *TwinsUK* consisted of human gut metagenomes from adults in the TwinsUK cohort [27], while the *Animal-gut* comprised gut metagenomes from a broad taxonomic diversity of vertebrates [28]. *Pinnell2019* and *Mantri2021* consisted of benthic and soil microbial community metagenomes, respectively. *MarineMetagenomeDB* and *TerrestrialMetagenomeDB* consisted of a selection of metagenomes from each respective database. The following filtering criteria were applied prior to metagenome selection: i) $\geq$ 1e6 and $\leq$ 80e6 reads, ii) maximum read lengths of $\leq$ 150bp, and iii) $\geq$ 10 samples in the study, and for the UHGG: iv) $\geq$ 10 and $\leq$ 300 MAGs associated with the metagenome.

We also evaluated ResMiCo on 2 mock community datasets: BMock12 [33] and MBARC-26 [34]. We subsampled each metagenome to 2M read pairs in order to evaluate a sequencing depth on par with existing real-world metagenomes [35, 36]. Representative genomes were downloaded from JGI IMG and Genbank for BMock12 and MBARC-26, respectively. These representatives were used in conjunction with MetaQUAST to identify true misassemblies.

Metagenome read data processing was done as described in Youngblut and colleagues [28]. Briefly, reads were validated with fqtools [37]. Adapters were trimmed with Skewer [38]. The "bbduk" command from bbtools (https://sourceforge.net/projects/bbmap/) was used to trim and filter reads based on Phred scores. The "bbmap" command from bbtools was used to filter reads mapping to the hg19 human genome assembly. Read quality reports for each step of the pipeline were generated and visualised with FastQC and MultiQC, respectively (https://www.bioinformatics.babraham.ac.uk/, [39]). Metagenomes were assembled via metaSPAdes with default parameters, and contigs < 1000bp were removed. We did not also assemble the metagenomes with MEGAHIT, given the extra computational expense and methods complexity of evaluating both assemblers for each dataset, along the improved accuracy of metaSPAdes versus MEGAHIT [3]. Read mapping and ResMiCo feature generation were conducted as done for the simulation datasets.

## Data preprocessing

Count features were normalized by coverage (the number of reads mapped to the position) such that they are in the 0–1 range. For numerical features, we pre-computed mean and standard deviations using all contigs in the *n9k-train* dataset and saved these values. For all datasetes, we standardized numerical features to set the mean to zero and the variance to one using values computed on the training set. Missing values were replaced by zero (the new mean). We summarize the preprocessing applied to each feature in Table A in S1 Text.

Since we observed that ResMiCo did not generalize well to insert size distributions substantially deviating from the training dataset (Table E in S1 Text), we excluded metagenomes for which the 0.05 and 0.95 quantiles of the mean insert size distribution lay outside the 0.02 and 0.98 quantiles of the mean insert size distribution of the *n9k-train* dataset, which are equal to 117 and 493, respectively.

## Model and training

**Architecture.**   The ResMiCo neural network (NN) architecture is shown in Fig 2. It belongs to a class of deep convolutional residual neural networks. Residual connections enable deeper (more layers) neural networks and more efficient training compared to convolutional NNs [20]. Deeper models can capture more complex patterns spread over larger inputs. The main building unit, depicted at the bottom of the figure, is the residual block, consisting of two batch-normalized-convolutions [40] with a ReLu activation. The input of the residual block is

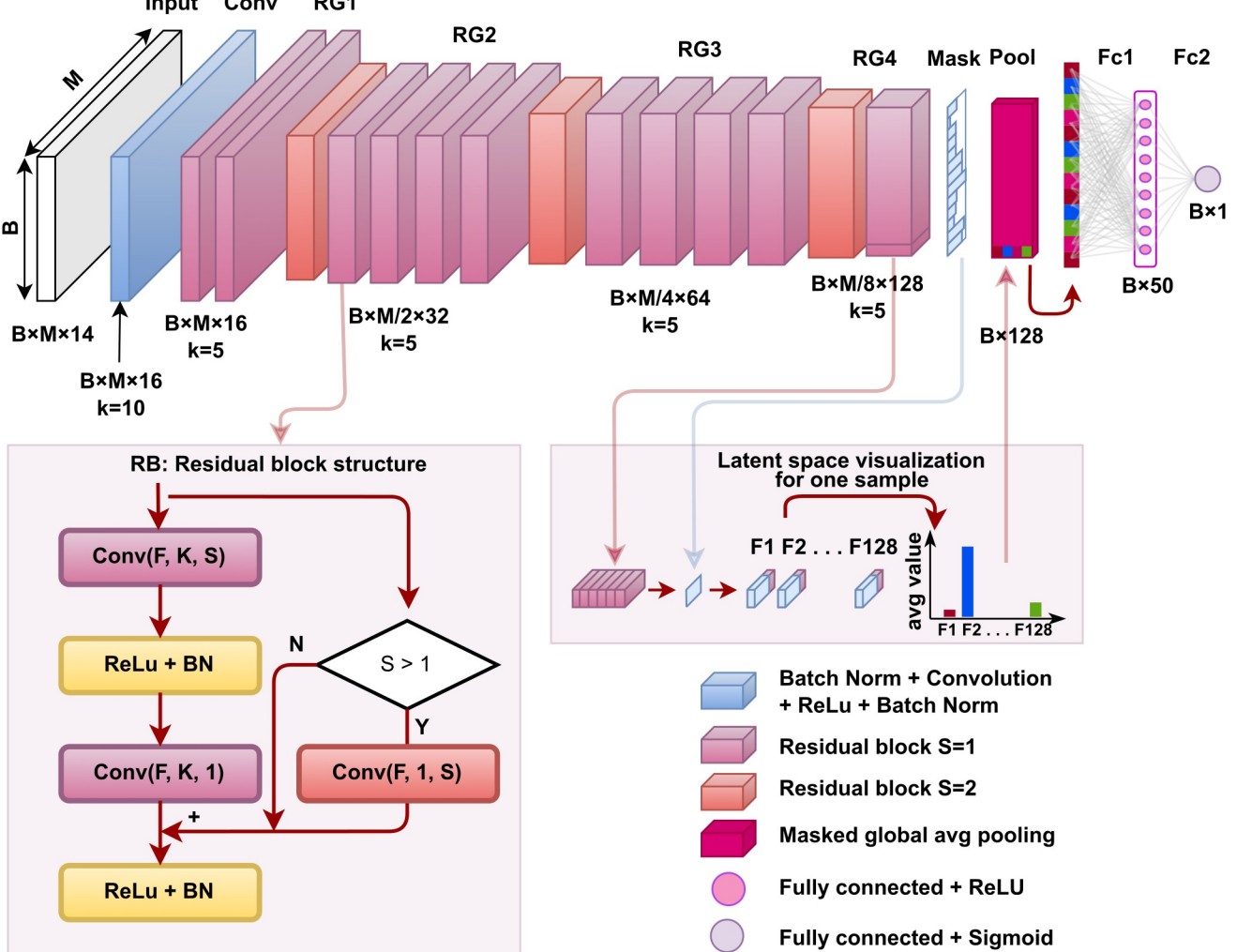

**Fig 2. ResMiCo architecture.** The input is first passed through multiple convolutional layers; then the convolved result is masked to eliminate the effect of padding and passed through an average pooling layer, followed by two fully connected layers of sizes $128 \times 50$ and $50 \times 1$. The convolutional part consists of a simple convolution, followed by four residual groups (RG) with 2, 5, 5, and 2 residual blocks, respectively. The bottom of the figure depicts the structure of a residual block with a given number of features (F), kernel size (K), and stride (S). The first convolution in RG2, RG3, and RG4 halves the input size (using a stride of $S = 2$) and doubles the number of filters, gradually from 16 to 128. B denotes the batch size, and M represents the maximum contig length. The "14" in "BxMx14" represents the number of the selected features in the input to the neural network. Overall, ResMiCo has 562,573 parameters, of which 559,441 are trainable.

connected with the output by simple element-wise addition. Since convolutions are not padded, the last $2*(K-1)$ positions in the residual input, where $K$ is the convolution kernel size, are cut off to match the output size. When the residual block is downsampling the data (by using a stride of $S = 2$ in the first convolution), the residual input is downsampled with a $K = 1$ convolution of an identical stride.

Residual blocks with the same number of filters and identical output shapes are grouped into *residual groups*. ResMiCo has 4 residual groups, with the center groups consisting of 5 residual blocks each, while the outside groups contain 2 residual blocks. Each of the last three residual groups starts with a convolution that doubles the number of filters and halves the input size using a stride $S = 2$. All layers use a ReLU activation, except for the last fully connected layer, which uses a sigmoid activation. The output of the convolutional layers is

summarized along the spatial axis via global average pooling, resulting in an output shape that depends only on the number of filters in the last convolutional layer (128 in our case) rather than on the contig length, thus allowing ResMiCo to handle contigs of variable length. Contigs in a batch are padded to the longest length, and the effects of padding are neutralized by creating a mask that is fed to the global average pooling layer. The resulting 128 features of the global average pooling are fed into the final two layers: a fully connected layer with 50 neurons and a one-neuron output layer with a sigmoid activation function.

**Training.**   The model was trained on the *n9k-train* training dataset for 50 epochs. One *epoch* is one pass of the entire training dataset through the algorithm. All misassembled contigs were used as positive training examples. In contrast, we randomly selected a 10% subset at every training epoch for the over-represented class of correctly assembled contigs, thus artificially increasing the positive sample rate to 24%. This helped to balance the dataset and reduce the computational load during training. For contigs shorter than 20,000 base-pairs, the entire contig is selected and zero-padded to the maximum batch length. For misassembled contigs longer than 20,000 base-pairs, a random 20,000 base-pair interval around each breakpoint (as identified by MetaQUAST) was selected. For long contigs with no misassemblies, a random 20,000 base-pair interval is selected.

During model training, the binary cross-entropy loss between the target and predicted output was minimized by an Adam optimizer [41]. We used a batch size of 200 and an initial learning rate of 0.0001 with exponential decay of 0.8 when plateauing at evaluation. Gradients were clipped to a norm of 1 and a value of 0.5.

**Model selection.**   We used 10% of the *n9k-train* dataset as a validation set for model selection (*n9k-valid*). AUPRC on the validation set was computed every second epoch; if the score improved, a corresponding model was saved. The ResMiCo model described in this section achieved the highest AUPRC on the *n9k-valid* dataset at Epoch 46. The list of optimized hyperparameters and the attempted values are provided in Table B in S1 Text.

**Feature selection.**   Since ResMiCo uses a larger number of features than both DeepMAsED and metaMIC, it is important to understand the amount that each feature, particularly features unique to ResMiCo, contributes to model predictions. Borrowed from game theory, Shapley values provide a principled way of explaining the predictions of machine learning models. We approximated the Shapley values using the Deep Shap (SHAP) algorithm [42], a refined version of DeepLIFT [43].

In order to be able to compute SHAP coefficients, we had to make some adjustments to ResMiCo's architecture: the input size was fixed, the padding was not masked, and the global average pooling layer was replaced by local pooling with a window covering the whole length. SHAP requires as input background samples as well as samples for which the predictions will be explained. We randomly sampled 200 contigs for the background and 200 contigs for explanations (100 correctly assembled and 100 misassembled) from the *n9k-novel* dataset. In Fig L in S1 Text, we show that SHAP led to the same conclusions when using subsets of the 200 contigs, suggesting 200 is sufficient for this analysis.

Fig 3 shows features ranked by their importance. For comparison, we also marked features present in the ResMiCo pipeline that were used by metaMIC and DeepMAsED. Feature names are explained in Table A in S1 Text. The top 14 features were selected as input to the ResMiCo model. We included at least one feature of each kind: mapping quality, alignment score, etc. Limiting the number of features resulted in significantly reduced training time.

**Predictions.**   To predict the misassembly probability for contigs longer than 20,000 base pairs, we split the contig into chunks of 20,000 base pairs, with a stride (overlap) of 500 bases in order to mitigate problems when the breakpoint is located at the very end of a chunk and two consecutive chunks individually appear as correctly assembled contigs. The prediction for

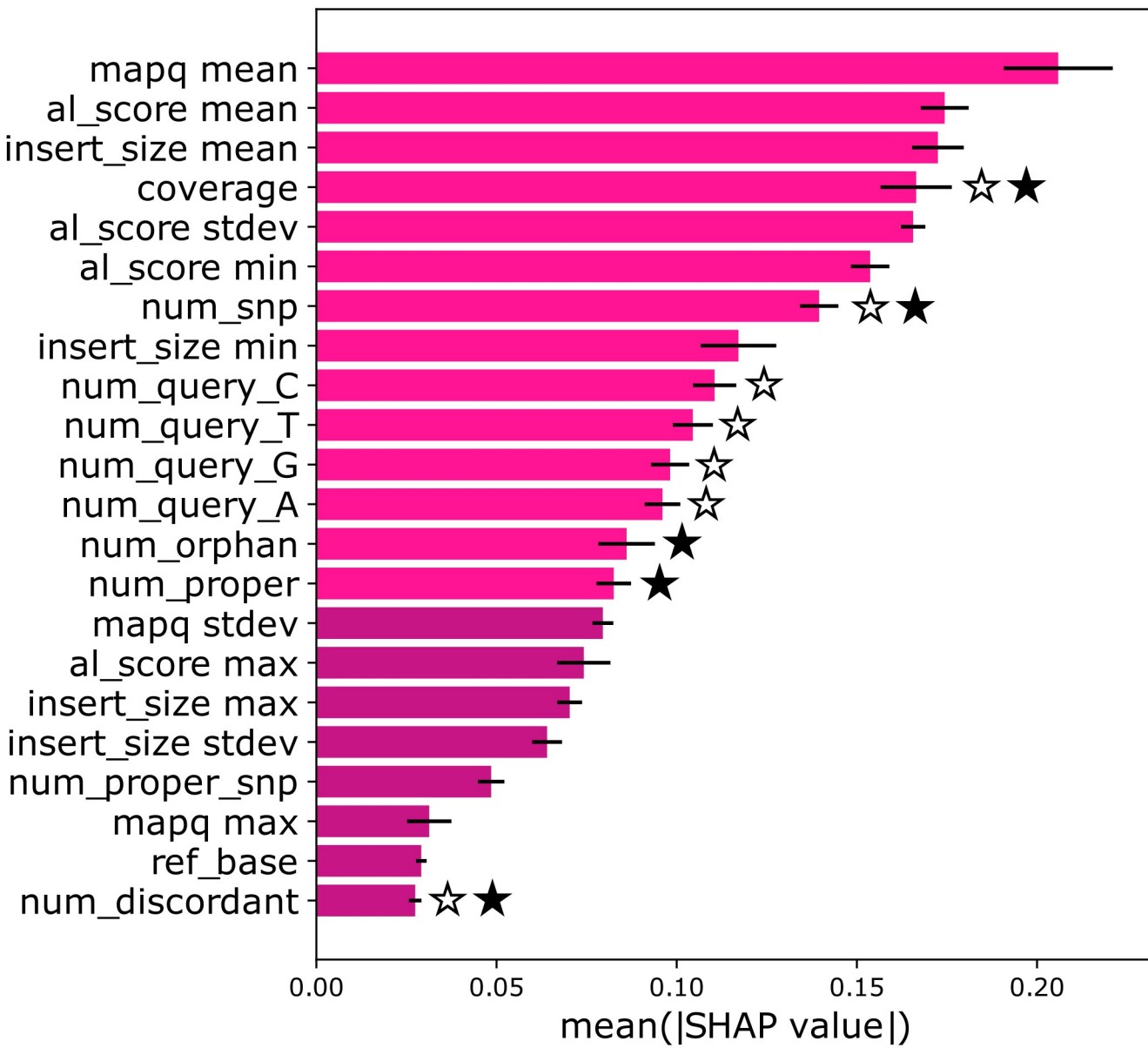

**Fig 3. Feature ranked by their importance.** The lighter color marks features used by the ResMiCo model. We denote features used by DeepMAsED and metaMIC with a star outline and filled star, respectively. `mapq` and `al_score` are mapping quality and alignment score, as defined by Bowtie2. `num_snp` is the number of SNVs among aligned reads relative to the reference. `num_query_[ATGC]` is the base composition of aligned reads at the target position. `num_orphan` is the number of aligned reads in which only one of the pairs aligns properly. `num_proper` is the number of read pairs that align properly, as defined by Bowtie2. `num_proper_snp` is properly aligned reads with a SNV relative to the reference at the target position. `ref_base` is the reference base `[ATGC]` at the target position. Error bars correspond to the stdev computed over 5 runs.

the contig was obtained by selecting the maximum score across all chunks. For contigs shorter than 20,000 base pairs, the entire contig is given as input.

## Comparison to the state of the art

We analyzed the performance of our proposed model in relation to the following existing methods:

- metaMIC [44]—applied with default parameters, except the minimum contig length was reduced from 5 kbp to 1 kbp, in order to test on the same data as the other methods;

- DeepMAsED [18]—we followed the feature generation scheme and the trained model provided by the authors;

- ALE [45]—we aggregated four positional sub-scores that ALE outputs (depth, place, insert, and k-mer log-likelihoods) with the same thresholds defined in [18]. The contig misassembly probability is computed as the number of positions with the sub-score below the threshold divided by contig length;

- Random—we assigned a random misassembly probability to each contig. This results in a horizontal line on a precision-recall curve plot with a precision equal to the prevalence of misassemblies in the dataset.

Since all datasets suffer from a class imbalance in the detriment of positive samples (misassembled contigs, Table 2), we selected the area under the precision-recall curve (AUPRC) as a metric to measure performance, rather than the area under the receiver operator curve (AUROC) [46]. However, AUPRC is not invariant to the prevalence of positive samples, so we used AUROC to compare the model's performance across datasets with different percentages of positive samples.

### Benchmarking ResMiCo resource requirements

We benchmarked ResMiCo with direct comparisons between utilizing one CPU versus one GPU. On the CAMI gut dataset, ResMiCo was > 2x faster with a GPU versus a CPU ($108 \pm 0.7$ versus $38.7 \pm 10.3$ contigs per second). While a GPU is substantially faster than a CPU, one could still process nearly 140,000 contigs in 1 hour with a single CPU. Nonetheless, we note that multiple GPUs are recommended for training the model on large datasets, given that training on CPUs would not be feasible for large datasets, as in this work.

## Results

### ResMiCo outperforms existing models and is robust to metagenome novelty

We first tested all models against the *n9k-novel* dataset, which consisted of family-level taxonomically novel genomes relative to any in the training dataset. ResMiCo outperformed DeepMAsED, ALE, and metaMIC by a large margin, with an AUPRC of 0.76 versus 0.25 for DeepMAsED, the second-best performing model (Fig 4A). Note that DeepMAsED's AUPRC score dropped from 0.57 (reported in [18]) to 0.25 due to a higher variability within the *n9k-novel* test set. Importantly, ResMiCo's AUPRC did not substantially differ between the training validation (0.73) and the *n9k-novel* dataset (0.76), thereby demonstrating that the model is robust to taxonomic novelty. ResMiCo's AUPRC score typically varied from 0.6 to 0.8 across the various simulation parameter combinations (Fig H in S1 Text). The most challenging settings for ResMiCo were a low community richness and a low sequencing depth. We also found that the contig length distribution explained much of the variability in ResMiCo model performance. For the simulations with a median contig length longer than 2000bp, the AUPRC was between 0.4 and 0.6 (Fig E in S1 Text). In terms of misassembly type, ResMiCo performance was lowest for inversions (Fig G in S1 Text).

We simulated another test dataset (*n2k-novel-intraspecies*) with more genomes per species ($33.3 \pm 115$) versus *n9k-novel* ($2.4 \pm 13.7$). ResMiCo performance on *n2k-novel-intraspecies* declined relative to *n9k-novel* (AUPRC = 0.487, AUROC = 0.955), which was likely due to an

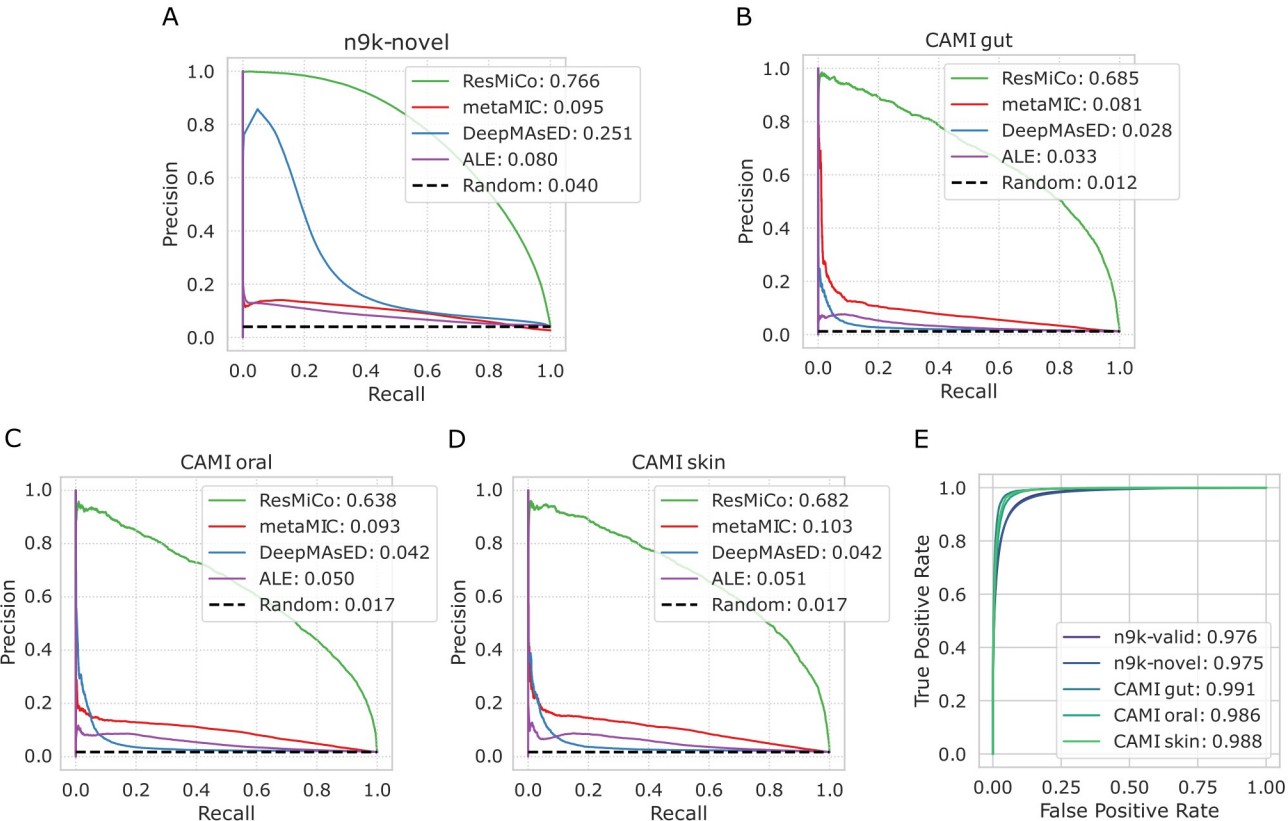

**Fig 4. ResMiCo performance evaluation.** Precision-recall curves and the corresponding AUPRC scores for ResMiCo and four baseline methods (metaMIC, DeepMAsED, ALE, Random) applied on the (A) *nk9-novel*, (B) CAMI gut, (C) CAMI oral, and (D) CAMI skin datasets. (E) Receiver operating characteristic curve and the corresponding AUROC scores for ResMiCo applied on five datasets: *n9k-train* (validation set only), *nk9-novel*, and three CAMI datasets.

increase in inter-genome translocations among closely related taxa. Still, ResMiCo greatly outperformed the closest competitor: metaMIC (AUPRC = 0.080, AUROC = 0.805).

We next evaluated ResMiCo on the CAMI gut, oral, and skin metagenome datasets, which are commonly used simulation datasets for the evaluation of metagenomics analysis tools. The CAMI datasets differed substantially from *n9k-train* and *n9k-novel* in regards to coverage (sequencing depth) and class imbalance (percent misassemblies) (Table 2). Moreover, reference genomes used for the *n9k-train* and *n9k-novel* datasets were selected from the entire GTDB, while the CAMI datasets consisted of biome-specific reference genomes [24]. Regardless of these differences, ResMiCo's performance remained largely unaffected, and the model still clearly outperformed all competitors (Fig 4B, 4C and 4D). Given that the 5 synthetic datasets differ substantially in true positive rates (Table 2), we computed the AUROC score, which is unaffected by such differences. Fig 4E shows that the AUROC remains relatively constant across the *n9k-train* validation, *n9k-novel*, and the CAMI datasets.

We also evaluated ResMiCo on two CAMI datasets that simulate non-human biomes: *CAMI-marine* and *CAMI-plant-associated*. The performance of ResMiCo was comparable to the human-associated CAMI datasets (*CAMI-marine*: AUPRC = 0.831, AUROC = 0.990; *CAMI-plant-associated*: AUPRC = 0.611, AUROC = 0.965), suggesting that ResMiCo can generalize to metagenomes from highly varying biomes.

**Table 3. Low-quality contig filtering.** Statistics before and after filtering low-quality contigs with the ResMiCo model applied on the *n9k-novel* test set. The ResMiCo score threshold was set to >0.8. Misassemblies length is the sum of misassembled contig lengths divided by the total number of bases.

| Dataset | True error rate | Misassemblies length | N contigs | N50 | Mean length | Median length |
|---------|----------------|---------------------|-----------|-----|-------------|---------------|
| Original | 4.0% | 4.3% | 6779977 | 3919 | 2911 | 1510 |
| ResMiCo | **1.2%** | **1.8%** | 6474434 | 3917 | 2914 | 1522 |

Lastly, we assessed 2 mock community datasets: BMock12 [33] and MBARC-26 [34]. While existing mock communities are orders of magnitude lower in richness than complex communities, such as soil or mammalian gut microbiomes, and usually completely lack intra-species genomic diversity, mock communities do provide the best available real-world, ground-truth datasets. ResMiCo performance was in line with our expectations based on our simulations of low community richness (Fig H and I in S1 Text), with an AUPRC of 0.475 and an AUROC of 0.951 when averaged across the 2 datasets (Table H in S1 Text). These findings demonstrate that ResMiCo can generalize well to real-world metagenomic data.

## Improvement of assembly quality after filtering ResMiCo-identified misassemblies

A primary function of ResMiCo is to identify misassembled contigs so that they can be removed from the assembly. To illustrate the effects of such filtering, we discarded contigs in *n9k-novel* with a ResMiCo score of >0.8, which corresponds to a recall and precision of 0.72 and 0.65, respectively. Given that truly misassemblied contigs are known for the *n9k-novel* dataset, we could measure the true error rate before and after filtering. Filtering according to ResMiCo scores resulted in a reduction of the true error rate from 4% to 1% while keeping the contiguity metrics virtually unmodified (Table 3).

We also evaluated whether the MetaQUAST-defined "genome fraction" (the percentage of aligned bases in the reference genome) substantially declined as a result of filtering ResMiCo-identified misassemblies. For both the *CAMI-gut* and *CAMI-marine* datasets, the genome fraction did not substantially change after filtering (Wilcox, $P \geq 0.66$; Fig K in S1 Text).

## Assembly optimization based on ResMiCo-identified error rates

Since ResMiCo generates a score for each contig, we could use the number of contigs with a score above a certain threshold to estimate the misassembly rate of a given contig set (we used >0.8 in our experiments). The estimated misassembly rate could then be used to optimize metagenome assembler parameters (e.g., k-mer lengths) for real metagenomes, which lack ground-truth data. Assembler hyperparameters are generally optimized simply based on total contiguity (e.g., N50) or possibly via CheckM after binning contigs into MAGs. However, such methods do not directly assess contig assembly accuracy. In order to use ResMiCo for this application, model performance must be robust to assembler hyperparameter settings outside of the training distribution.

We tested ResMiCo's performance as an oracle for assembler performance by simulating datasets in a similar fashion as *n9k-novel*, but with 6 different k-mer length hyperparameter settings for both MEGAHIT and metaSPAdes (see Methods). For each of the 6 k-mer length combinations, we generated akin to *n9k-novel* but utilized only 2 community richness (50 and 3000) and 2 sequencing depth (2M and 8M) settings. The rest of the simulation parameters were fixed: genome abundance distribution with $\sigma = 1$, read lengths of 150 bps, insert size distribution of mean = 270 & sd = 50, and the "HiSeq 2500 error" error profile. The percentage of actual misassembled contigs differed from <1% to 30% depending on the assembler and the

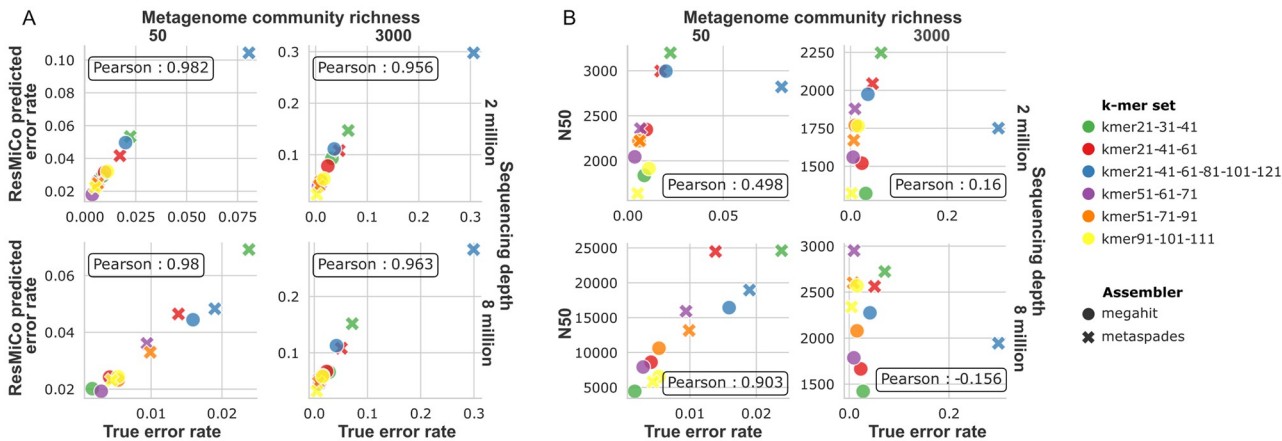

**Fig 5. Misassembly (error) rate produced by MEGAHIT and metaSPAdes assemblers with six different k-mer sets.** k-mer set names denote the k-mer lengths used for the assembly. (A) ResMiCo-identified error rate (y-axis) correlates with the true error rate. (B) N50 size and the true error rate are orthogonal measures of the metagenome quality.

chosen k-mer set applied on the same set reads (Fig 5). We then compared the percentage of misassembled contigs with the percentage estimated by ResMiCo for each of the four community richness/sequencing depth combinations (6 k-mer sets per combination).

ResMiCo was able to accurately rank the assemblies by misassembly rate for all four simulation parameter combinations, achieving a Pearson correlation of 0.9. (Fig 5A). While ResMiCo has a tendency to overestimate the misassembly rate with a selected prediction threshold, and the ratio between predicted and true misassembly rate depends on sample richness and sequencing depth (Fig J in S1 Text), the ranking remained consistent in all considered scenarios. At the same time, for the most well-assembled metagenomes (low richness and high sequencing depth), we observed a correlation of 0.9 between N50 and true error rate (Fig 5B), which suggests that the high contiguity achieved together with high misassembly error rate. However this relationship does not hold for the samples simulated with other parameters, making possible to search for assembler parameters that produce good quality in terms of contiguity and error rate simultaneously. Consequently, we propose that ResMiCo can be used to rank assembler parameters for real-world metagenome data and identify parameters leading to the lowest misassembly rate.

## Latent space visualization

To get an intuition on how ResMiCo internally represents data, we studied the output of the global average pooling layer. At that point, the input data is mapped into a 128-dimensional space. We used UMAP [49] with default parameters to project the embeddings into a two-dimensional space. UMAP was fitted on the *n9k-train*, *n9k-novel*, and *CAMI-gut* datasets. We used 10,000 randomly sampled contigs from each of the three datasets.

The latent space visualization indicates that *n9k-train* has more variability (due to the extensive set of parameters used in the simulations) than *CAMI-gut*, which is concentrated in a small subspace, while misassembled contigs from both datasets are generally clustered together (Fig 6A and 6B). Note that since ResMiCo has two fully connected layers following the visualized global average pooling, the two classes are not expected to be completely separable at this stage. Both community richness and average contig coverage strongly partition the latent space (Fig 6C and 6D).

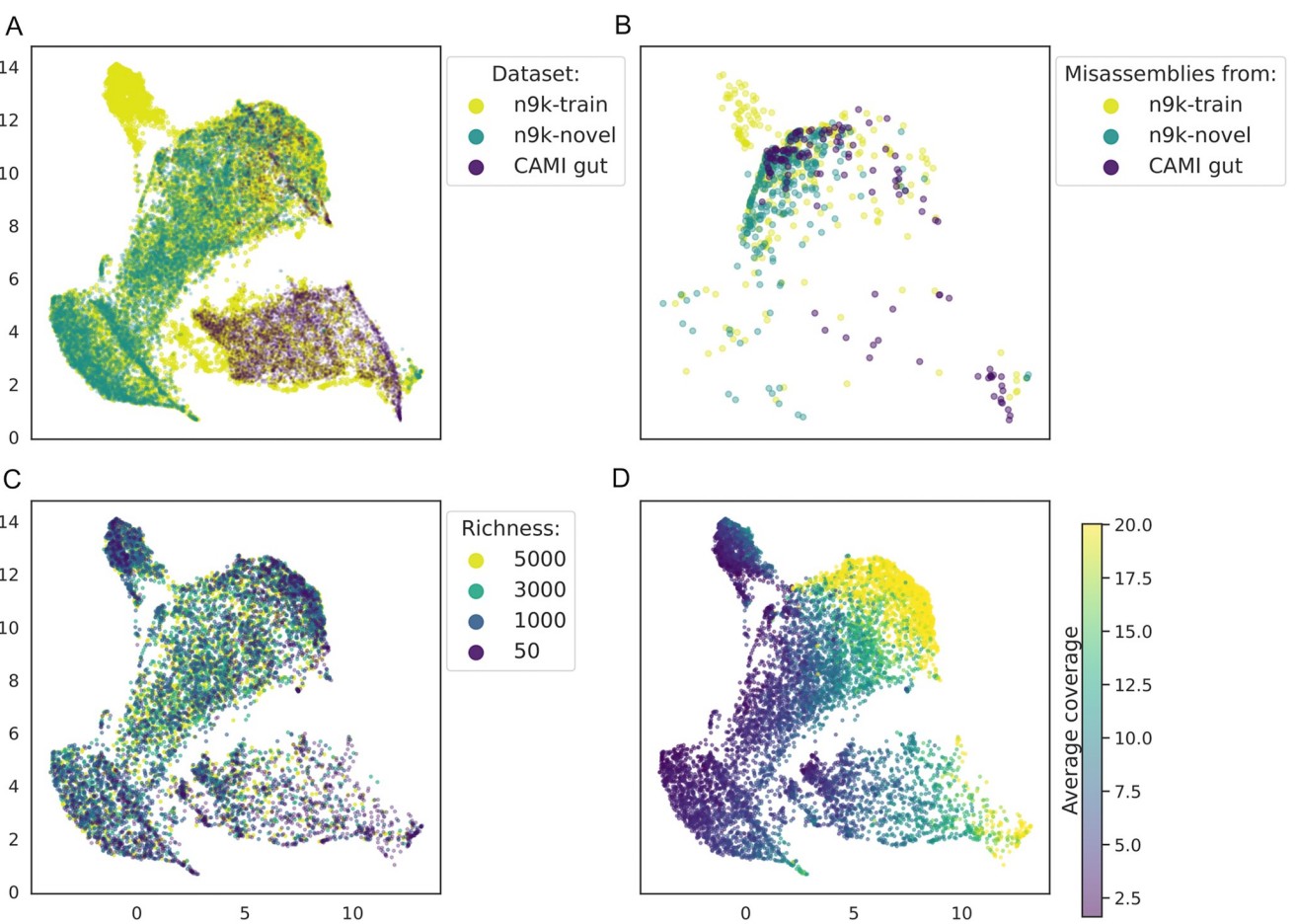

**Fig 6. Contig embeddings learned by ResMiCo, projected using UMAP.** The first row shows (A) all contig embeddings and (B) misassembled contig embeddings for the *n9k-train*, the *n9k-novel*, and the *CAMI. gut* datasets. In the second row, contigs from the *n9k-train* dataset are colored (C) by the richness of the simulated community they originated from and (D) by their average coverage.

### ResMiCo detects a 3–12% misassembly rate in real-world metagenomes

We applied ResMiCo to published gut metagenome datasets from multiple studies in order to assess the prevalence of misassembled contigs in publicly available metagenomic data. We utilized three datasets of gut samples: *UHGG*, *TwinsUK*, and *Animal-gut*. *UHGG* consisted of a random subset of gut metagenomes associated with MAGs in the UHGG database, while *TwinsUK* and *Animal-gut* consisted of gut metagenomes from westernized adults and a broad taxonomic diversity of vertebrates, respectively (see Materials and methods). We also utilized 2 marine meatgenome datasets (*Pinnell2019* and *MarineMetagenomeDB*) and 2 from soil (*Mantri2021* and *TerrestrialMetagenomeDB*).

ResMiCo detected an average of 3.4%, 6%, and 8.8% misassembled contigs across all metagenome assemblies in the vertebrate-associated datasets: *TwinsUK*, *Animal-gut*, and *UHGG*, respectively. The marine and soil datasets contained more misassemblies (*MarineMetagenomeDB*: 8.3%, *Mantri2021*: 11.5%, *Pinnell2019*: 8.6%, and *TerrestrialMetagenomeDB*: 12.5%). Overall, we evaluated 8,235,502 contigs, of which 6% are misassembled according to ResMiCo's predictions ($\geq 0.8$).

We provide an estimated the misassembly rate for each metagenome in S1 Table. Averaged across all metagenomes, the misasssembly rate was 7% ± 5.7 (sd). The high variability among

samples and datasets suggests that sample-specific factors (e.g., community taxonomic complexity or variability among NGS library preparations) can substantially influence misassembly rates. Note that we did not apply ResMiCo on the minority of metagenomes that failed to meet our inclusion criteria (see the Materials and methods).

## Discussion

We addressed the problem of reference-free metagenome quality evaluation by developing ResMiCo, a deep residual neural network that enables accurate misassembled contig identification. ResMiCo provides an efficient data generation pipeline (see Text G in S1 Text), which transforms raw reads and contigs into positional features that are utilized by a residual neural network to predict if a given contig was misassembled. The ResMiCo model was trained and tested on datasets of unprecedented size and complexity (*n9k-train* and *n9k-novel* contain 210 Gbps of assembled contigs), which we have made freely available as a resource for further model development and benchmarking (see Materials and methods). These datasets can be expanded, or new datasets can be generated with ResMiCo's dataset simulation pipeline, which allows for straightforward, efficient data generation on high-performance computing systems.

ResMiCo achieved a 0.76 AUPRC score on the taxonomically novel holdout test set (*n9k-novel*), which is an exceptional improvement over the state of the art (Fig 4A). The robustness of ResMiCo to family-level taxonomic novelty clearly demonstrates that it can be applied to metagenomes that include substantial taxonomic novelty, such as gut microbiomes from poorly studied animal species [28].

When tested on the 3 human and 2 non-human CAMI datasets, ResMiCo showed similarly high performance and again substantially outperformed the state of the art (see Results and Fig 4B, 4C and 4D). These results show that ResMiCo can generalize to third-party, biome-specific datasets, despite our use of a biome-agnostic training dataset consisting of genomes randomly selected from the entire GTDB.

ResMiCo is primarily designed to increase the quality of existing assemblies, and we demonstrated that filtering out contigs with high scores from the holdout test set resulted in a four-fold decrease in the true error rate without a substantial decline in contiguity (Table 3).

When applied to the real-world metagenome datasets, ResMiCo detected 7% ± 5.7 (sd) misassembled contigs per metagenome. This estimate is substantially higher than the 1% misassembly rate previously estimated via DeepMAsED [18], which may be due to differences in model accuracy and the increased number and variety of real-world metagenomes used for our estimation.

We also show that ResMiCo can be applied to select assembler parameters corresponding to the best assembly accuracy for a given unlabeled dataset (Fig 5). Researchers can thus optimize assembler parameters for obtaining high accuracy on their specific real-world metagenomes without relying on benchmarks from simulated datasets [24].

ResMiCo's vastly improved performance relative to other reference-free misassembly detection methods is likely due to three main factors. First, ResMiCo was trained on a very large and varied dataset. Even after re-implementing the *samtools pileup* algorithm to gain a 10x speed improvement, generating the *n9k-train* dataset required nearly 40,000 CPU hours to produce all 4,560 simulation parameter combinations (Table 1). For model training, we used 52.5M contigs that total 159 Gbps. In contrast, the DeepMAsED training dataset is 100x smaller, while no such training dataset information is available for metaMIC [44]. The UMAP projections of the contig embeddings for the *n9k-train*, the *n9k-novel*, and the *CAMI-gut* datasets (Fig 6A) show that ResMiCo's training data comprises a substantial portion of the input space. Second,

ResMiCo was trained on a larger number of carefully selected features; although ResMiCo uses only the top 14 features in Fig 3, we generated and tested a total of 23 features (Table A in S1 Text) before selecting the 14 best performing ones. In comparison, DeepMAsED used 8 features for training, while metaMIC used 4 types of features, and as we have shown, both models failed to identify the features most relevant to misassembly detection. Third, introducing residual blocks, combined with the larger dataset, allowed us to train a deeper convolutional model, which has been shown to have better performance relative to traditional, shallower CNNs [20]. Long range (up to 20,000 bp) signal from raw positional features is transformed by the residual blocks, such that ResMiCo is able to internally detect a breakpoint (Fig B in S1 Text) and identify misassembled contigs based on the strength of this signal.

While our extensive evaluations showed that ResMiCo is robust to many sources of dataset novelty, we did find that the model is sensitive to the mean insert size distribution (Table E in S1 Text). Therefore, prior to evaluation, we removed all metagenomes falling substantially outside of the *n9k-train* insert size distribution. While filtering by insert size may lead to a biased selection of real-world metagenomes, Illumina sequencing libraries often vary substantially in fragment length and subsequently insert size distribution, even within the same sequencing run. Such variation can result from inaccuracies in DNA quantification and reagent aliquoting. ResMiCo automatically detects if the insert size distributions of the evaluated data differ substantially from those during training and warns users that results may be less accurate. If needed, researchers can simulate more training data that includes the desired insert size range with the ResMiCo simulation pipeline.

Improving performance robustness across changing distributions, sometimes termed out-of-distribution (o.o.d.) generalization [50–52], could be an interesting direction for follow-up work. While deep learning has produced impressive results in a range of domains, its reliance on independent and identically distributed (i.i.d.) training data can be a problem [53–55]. Although a range of attempts exist to improve o.o.d. generalization, empirical risk minimization still is the method of choice in practice [56, 57], especially when used with a dataset of maximal diversity, as done in our study.

Besides improving o.o.d., there are some other areas for further improvement. First, more research is needed to evaluate the quality of contigs assembled from error-prone long reads (e.g., Oxford Nanopore). Second, it is worth investigating if ResMiCo can be adapted to indicate the location of breakpoints in misassembled contigs. Third, rather than using binary labels, ResMiCo could be trained on misassembly type (e.g., inversion or translocation) to provide more detailed predictions. Fourth, while we did train on a very broad selection of genomes from across the bacterial and archaeal tree of life, future training and evaluation could be expanded to include eukaryotes and viruses.

In summary, ResMiCo is a major advancement in the challenge of reference-free metagenome quality assessment. Existing methods addressing this problem have not been widely used, likely due to concerns regarding whether such approaches can generalize to real-world datasets. Our extensive testing shows that ResMiCo generalizes well across a large parameter space that includes taxonomy, community abundances, and many sequencing parameters. Wide adoption of ResMiCo could substantially improve metagenome assembly quality for individual studies and databases, which is critical for obtaining accurate biological insights from metagenomic data.

## Supporting information

**S1 Text.** Fig A. Breakpoint locations for misassemblies identified by MetaQUAST (treated as ground-truth in this work). Fig B. Feature maps for four *n9k-novel* misassembled contigs. Fig

C. Feature maps from the last layer before global pooling for four *n9k-novel* correctly assembled contigs. Fig D. The contig length distribution and ResMiCo performance on contigs of different lengths. Fig E. AUPRC that ResMiCo achieves on 240 subsets (differing simulation parameters) of the *n9k-novel*. Fig F. Distribution of errors found by MetaQUAST in the *n9k-novel* test dataset. Fig G. ResMiCo scores for misassembled contigs grouped by assembly error type. Fig H. ResMiCo performance measured by AUPRC on contigs from the datasets with various simulation parameters (*n9k-novel*). Fig I. ResMiCo performance measured by AUROC on contigs from the datasets with various simulation parameters (*n9k-novel*). Fig J. Number of misassemblies found by ResMiCo divided by the true number of misassemblies in the datasets with various simulation parameters (*n9k-novel*). Fig K. Genome fraction measured on (A) CAMI gut and (B) CAMI marine datasets before and after filtering misassembled according to ResMiCo contigs. Fig L. Features ranked by their importance. Table A. The full list of positional features computed by ResMiCo pipeline. Table B. Hyperparameters tested for ResMiCo architecture. Table C. Effect of down sampling reads on the ResMiCo predictions. Table D. Contig length cut-off effect on the ResMiCo predictions. Table E. ResMiCo performance on test data varying by the *mean* of the insert size distribution. Table F. ResMiCo performance on the test sets with variable *stdev* of the insert size distribution. Table G. The insert size distribution statistics across synthetic and real-world datasets used in this work. Table H. ResMiCo performance on two mock real-world datasets. Text A: ResMiCo's embeddings highlight breakpoint locations. Text B: ResMiCo performance by assembly error type. Text C: NN achitecture selection. Text D: Effect of the length cut-off and the read down-sampling on ResMiCo predictions. Text E: ResMiCo sensitivity to changes in insert size distribution. Text F: Clustering of misassemblies. Text G: Optimizing data generation performance.
(PDF)

**S1 Table. ResMiCo estimates the prevalence of misassembled contigs in publicly available metagenomic data.**
(XLSX)

**S2 Table. Characteristics of clusters of misassembled contigs from the *n9k-novel* dataset.**
See more details about clustering in Text F in S1 Text.
(XLSX)

## Author Contributions

**Conceptualization:** Olga Mineeva, Nicholas D. Youngblut.

**Data curation:** Olga Mineeva, Daniel Danciu, Nicholas D. Youngblut.

**Formal analysis:** Olga Mineeva, Daniel Danciu, Nicholas D. Youngblut.

**Funding acquisition:** Bernhard Schölkopf, Ruth E. Ley, Gunnar Rätsch.

**Investigation:** Olga Mineeva, Daniel Danciu, Nicholas D. Youngblut.

**Methodology:** Olga Mineeva, Daniel Danciu, Nicholas D. Youngblut.

**Project administration:** Bernhard Schölkopf, Nicholas D. Youngblut.

**Resources:** Ruth E. Ley, Gunnar Rätsch.

**Software:** Olga Mineeva, Daniel Danciu, Nicholas D. Youngblut.

**Supervision:** Bernhard Schölkopf, Gunnar Rätsch.

**Validation:** Olga Mineeva, Daniel Danciu, Nicholas D. Youngblut.

**Visualization:** Olga Mineeva, Daniel Danciu, Nicholas D. Youngblut.

**Writing – original draft:** Olga Mineeva, Daniel Danciu, Nicholas D. Youngblut.

**Writing – review & editing:** Olga Mineeva, Daniel Danciu, Bernhard Schölkopf, Gunnar Rätsch, Nicholas D. Youngblut.

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
