## [Decision Letter · Decision Letter 0]

24 Aug 2022

Dear Dr. Youngblut,

Thank you very much for submitting your manuscript "ResMiCo: increasing the quality of metagenome-assembled genomes with deep learning" for consideration at PLOS Computational Biology.

As with all papers reviewed by the journal, your manuscript was reviewed by members of the editorial board and by several independent reviewers. In light of the reviews (below this email), we would like to invite the resubmission of a significantly-revised version that takes into account the reviewers' comments.

The reviewers all agree that the methods are an improvement over the state-of-the-art and that the tool could potentially be widely used, but raise several important points related to both unclear presentation and whether some of the conclusions are sufficiently supported. Of particular interest to us are the concerns related to the evaluation on real data: the suggestion of reviewer #3 to test ResMiCo on data with a known standard seems very reasonable; similarly, the comment of reviewer #1 that only human gut samples are used should be addressed (ideally by including additional benchmarks).

We cannot make any decision about publication until we have seen the revised manuscript and your response to the reviewers' comments. Your revised manuscript is also likely to be sent to reviewers for further evaluation.

Sincerely,

Luis Pedro Coelho

Academic Editor

PLOS Computational Biology

William Noble

Section Editor

PLOS Computational Biology

The reviewers all agree that the methods are an improvement over the state-of-the-art and that the tool could potentially be widely used, but raise several important points related to both unclear presentation and whether some of the conclusions are sufficiently supported. Of particular interest to us are the concerns related to the evaluation on real data: the suggestion of reviewer #3 to test ResMiCo on data with a known standard seems very reasonable; similarly, the comment of reviewer #1 that only human gut samples are used should be addressed (ideally by including additional benchmarks).

Reviewer's Responses to Questions

**Comments to the Authors:**

Reviewer #1: The review is uploaded as an attachment.

Reviewer #2: This paper addresses the major challenge of evaluating contigs misassembly from the metagenomic reads without reference requirements. The proposed system, ResMico, utilizes a novel deep convolutional neural network with skip connections between non-adjacent layers. The authors have performed extensive evaluations through simulated datasets representing variously plausible microbial communities and sequencing parameters utilizing reference bacterial and archaeal genomes from the Genome Taxonomy Database. They showed the superior performance (accuracy and robustness) of ResMico compared to the existing methods, including DeepMAsDB, the tool previously developed by the authors’ group. They further showcased ResMiCo’s ability to detect an average of 4.7% error rate of contigs/metagenome in a large collection of gut metagenomes. The paper presents a tool with significant utility in metagenomics research. It is generally well-written; however, some technical aspects could be better explained.

1) The Residual block structure was used in ResMiCo. However, more detailed descriptions are needed to clarify the rationale for using it, and how the number of RGs was determined. Please also indicate in the Fig.2 caption that 14 represents the number of the selected features in the input to the neural network.

2) P9: Clarify why MetaSPAdes (but not MEGAHIT) was used for assembling the published, real-world metagenomes.

3) P12: the number of contigs used in the feature importance evaluation seems relatively small: 200 contigs randomly selected from the correct and incorrect. There is a concern if the size is sufficiently large for a good assessment of the importance. Could you discuss further on this?

4) P15: Using UMAP to examine how data were represented by ResMiCo is a good idea. However, as we know, UMAP only allows a visual inspection. I wonder if clustering the misassembled contigs would generate additional insights on the type of misassemblies in the data and if ResMiCo makes more mistakes in classification for any kind of misassembles represented by some cluster.

Reviewer #3: In this paper, the authors present ResMiCo, a deep convolutional neural network that is trained to identify misassembled contigs in a reference-free manner, from metagenome assemblies. While metagenome assemblies are rapidly accumulating, the quality of the assemblies can still be much improved in many cases. In this respect, it is critical to identify and correct misassembled contigs from existing metagenomes. As per results, ResMiCo outperforms prior methods *greatly*. While this is very positive, we have remained not yet fully convinced in terms of the experiments that support the idea of superiority. In the following some major comments.

* In lines 285-286, ”This filtering resulted in a reduction of the true error rate from 4% to 1% while keeping the contiguity metrics virtually unmodified.” How exactly do you estimate the error rate?

* In Metaquast, how does Genome fraction change after filtering out misassembled contigs?

* It would be clearly favorable to evaluate the method on real data that have a standard reference. Suggestions are: the Bmock12 data set in Volkan Sevim’s work treating shotgun metagenome data reflecting a mock community, sequenced using Oxford Nanopore, PacBio and Illumina technologies. Another dataset is the NWCs (natural whey culture) data set in Vincent Somerville’s “Long-read based de novo assembly of low-complexity metagenome samples results in finished genomes and reveals insights into strain diversity and an active phage system.” In both works, short reads and standard references are provided, where the latter was generated using hybrid assembly methods.

* Please provide Genome Fraction as an evaluation category generated by Metaquast. Since metagenomes often contain various strains referring to identical species, it is important to have a look at how Genome Fraction is affected. For example, loosing contigs from particular strains -- indicated by a drop in Genome Fraction -- would imply undesired biases.

**Have the authors made all data and (if applicable) computational code underlying the findings in their manuscript fully available?**

Reviewer #1: Yes

Reviewer #2: Yes

Reviewer #3: Yes

PLOS authors have the option to publish the peer review history of their article (what does this mean?). If published, this will include your full peer review and any attached files.

Reviewer #1: No

Reviewer #2: No

Reviewer #3: No
---

## [Decision Letter · Decision Letter 1]

9 Jan 2023

Dear Dr. Youngblut,

Thank you very much for submitting your manuscript "ResMiCo: increasing the quality of metagenome-assembled genomes with deep learning" for consideration at PLOS Computational Biology.

As with all papers reviewed by the journal, your manuscript was reviewed by members of the editorial board and by several independent reviewers. In light of the reviews (below this email), we would like to invite the resubmission of a significantly-revised version that takes into account the reviewers' comments.

We ask that the authors address the remaining concerns of reviewer #3.

We cannot make any decision about publication until we have seen the revised manuscript and your response to the reviewers' comments. Your revised manuscript is also likely to be sent to reviewers for further evaluation.

Sincerely,

Luis Pedro Coelho

Academic Editor

PLOS Computational Biology

William Noble

Section Editor

PLOS Computational Biology

We ask that the authors address the remaining concerns of reviewer #3.

Reviewer's Responses to Questions

**Comments to the Authors:**

Reviewer #1: Thanks to the authors for addressing all of my concerns and comments from the first review in the revised manuscript. I have no remaining questions and wish the tool will be well maintained and developed continuously in the future.

Reviewer #2: All my comments and suggestions have been satisfactorily addressed.

Reviewer #3: I disagree with the authors on not running the method on mock communities. While these mock communities are not *real-world* data, they are *real* data, and provide -- as widely accepted by the community -- a basic test that a method does not perform in an erratic manner when being confronted with real data. (thanks for listing the communities, but I am aware of everything you mention in your response anyway)

So, please run your method on mock communities, or convince me otherwise that it does not behave in mistaken ways when confronted with *real* data. (there was no mentioning in the response that you had done this; and I sometimes had the feeling that there was some confusion about the difference between *real-world* and *real* data).

Thank you very much.

**Have the authors made all data and (if applicable) computational code underlying the findings in their manuscript fully available?**

Reviewer #1: Yes

Reviewer #2: Yes

Reviewer #3: Yes

PLOS authors have the option to publish the peer review history of their article (what does this mean?). If published, this will include your full peer review and any attached files.

Reviewer #1: No

Reviewer #2: No

Reviewer #3: No
---

## [Editor Report · Decision Letter 2]

6 Mar 2023

Dear Dr. Youngblut,

We are pleased to inform you that your manuscript 'ResMiCo: increasing the quality of metagenome-assembled genomes with deep learning' has been provisionally accepted for publication in PLOS Computational Biology.

Best regards,

Luis Pedro Coelho

Academic Editor

PLOS Computational Biology

William Noble

Section Editor

PLOS Computational Biology

---

## [Editor Report · Acceptance letter]

19 Apr 2023

PCOMPBIOL-D-22-00907R2 

ResMiCo: increasing the quality of metagenome-assembled genomes with deep learning

Dear Dr Youngblut,

I am pleased to inform you that your manuscript has been formally accepted for publication in PLOS Computational Biology. Your manuscript is now with our production department and you will be notified of the publication date in due course.

With kind regards,

Anita Estes
